# CryptoMoE: Privacy-Preserving and Scalable Mixture of Experts Inference via Balanced Expert Routing

**Yifan Zhou**[†]
Peking University

**Tianshi Xu**[†]
Peking University

**Jue Hong**
Independent Researcher

**Ye Wu**
Independent Researcher

**Meng Li**[*]
Peking University

## Abstract

Private large language model (LLM) inference based on cryptographic primitives offers a promising path towards privacy-preserving deep learning. However, existing frameworks only support dense LLMs like LLaMA-1 and struggle to scale to mixture-of-experts (MoE) architectures. The key challenge comes from securely evaluating the dynamic routing mechanism in MoE layers, which may reveal sensitive input information if not fully protected. In this paper, we propose CryptoMoE, the first framework that enables private, efficient, and accurate inference for MoE-based models. CryptoMoE balances expert loads to protect expert routing information and proposes novel protocols for secure expert dispatch and combine. CryptoMoE also develops a confidence-aware token selection strategy and a batch matrix multiplication protocol to improve accuracy and efficiency further. Extensive experiments on DeepSeekMoE-16.4B, OLMoE-6.9B, and QWenMoE-14.3B show that CryptoMoE achieves $2.8 \sim 3.5\times$ end-to-end latency reduction and $2.9 \sim 4.3\times$ communication reduction over a dense baseline with minimum accuracy loss. We also adapt CipherPrune (ICLR'25) for MoE inference and demonstrate CryptoMoE can reduce the communication by up to $4.3\times$. Code is available at: https://github.com/PKU-SEC-Lab/CryptoMoE.

## 1 Introduction

Sparsely-gated mixture-of-expert (MoE) models have emerged as a powerful architecture for scaling up large language model (LLM) capacity without proportionally increasing the computation cost. As a result, many state-of-the-art LLM families, including LLaMA-4 [1], DeepSeek-V3 [2], and QWen-3 [3], have adopted MoE as their core architecture.

Driven by the high model capacity, MoE-based LLMs are increasingly adopted in real-world applications, some of which involve sensitive user data, e.g., person re-identification [4] and medical diagnostics [5]. Therefore, data privacy has become a major concern and has propelled the development of privacy-preserving inference frameworks. Hybrid cryptographic approaches combining Homomorphic Encryption (HE) and Secure Multi-Party Computation (MPC) are considered a promising solution. They enable the user and model provider (server) to jointly compute LLM outputs without exposing either the user inputs or the model weights.

However, existing private inference frameworks primarily support dense architectures such as GPT-2 [6] and LLaMA-1 [7], and lack support for MoE-based models. A core challenge lies in **how to**

---

[†]These authors contributed equally to this work.
[*]Corresponding author: meng.li@pku.edu.cn

39th Conference on Neural Information Processing Systems (NeurIPS 2025).

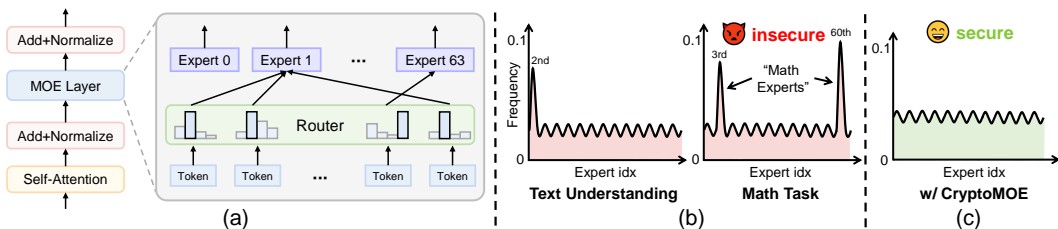

Figure 1: (a) Structure of MoE-based LLM; (b) Expert activation in 10-th layer of DeepSeekMoE differs notably between text understanding and math reasoning tasks. The underlying data is provided in Appendix A; (c) CryptoMoE features a privacy-preserving balanced expert routing.

**securely evaluate the dynamic routing mechanism inherent to MoE layers.** As illustrated in Figure 1(a), MoE operates by activating a subset of experts for each input token, where each expert is a distinct sub-network. Figure 1(b) further demonstrates that expert activation patterns are highly input-dependent and need to be protected: in 10-th layer of DeepSeekMoE [8], experts #3 and #60 are frequently activated for mathematical tasks across two math datasets [9, 10], but exhibit a uniform distribution across eight textual reasoning datasets [11, 12, 13, 14, 15, 16, 17]. Similar findings have been reported in previous work [18, 19]. This indicates that individual experts often specialize in specific semantic domains. Consequently, revealing expert routing information may leak sensitive details about both the input type and the internal specialization of different experts.

A natural solution to protect the routing information is to eliminate the sparsity of MoE and route all tokens through all experts, denoted as the dense baseline. However, this approach protects privacy at the cost of significant computation. For instance, in QWenMoE (4 out of 60 experts), it increases computation by about $15\times$, eliminating the efficiency benefits of the MoE structure.

To this end, we propose CryptoMoE, the first framework enabling private, efficient, and accurate inference for MoE-based LLMs. CryptoMoE features a key idea we term **Inference-Time Balanced Expert Routing**. Each expert processes a fixed number of tokens, denoted by $t$, regardless of the actual routing results. Tokens beyond this limit are discarded, making expert contributions input-independent and preserving privacy. By carefully selecting $t$, we can achieve strong privacy guarantees with little or no increase in overall computation.

However, naively discarding tokens exceeding the threshold $t$ leads to significant accuracy degradation. To alleviate this, we introduce a confidence-aware selection strategy. Among the tokens assigned to a given expert, we re-rank them by their routing confidence and retain only the top-$t$ tokens. To further support private inference under the balanced expert routing, we design a confidence-aware secure dispatch protocol that privately assigns tokens to their target experts, ensuring each expert receives the top-$t$ tokens with the highest routing probabilities. A corresponding secure combine protocol aggregates the expert outputs and reconstructs the final result for each token. Together, these protocols introduce only around 18% additional communication and computation overhead, while reducing expert computation by $8 \sim 15\times$, without leaking any routing information.

We further identify the expert linear layers as the main computational bottleneck in MoE inference. To address this, we propose an efficient Batch Ciphertext-Plaintext Matrix Multiplication (Batch MatMul) protocol, which packs tokens assigned to different experts into a single ciphertext. This reduces the number of costly HE rotation operations by a factor of $n$, where $n$ is the number of experts, significantly improving inference efficiency.

We evaluate CryptoMoE on three representative MoE-based LLMs: DeepSeekMoE-2.8B/16.4B[8], QWenMoE-2.7B/14.3B [20], and OLMoE-1.3B/6.9B [21], across eight zero-shot reasoning tasks. Results show that CryptoMoE retains 99.2% of the original accuracy on average, while achieving a $2.8 \sim 3.5\times$ speedup over dense baseline. Moreover, CryptoMoE achieves efficiency comparable to the insecure baseline that fully reveals routing information. We also adapt CipherPrune [22]'s pruning protocol to the MoE setting and construct a strong baseline. Compared to it, CryptoMoE achieves up to $4.3\times$ communication reduction and $2.4\times$ latency reduction. These results establish CryptoMoE as the first framework to enable private, efficient, and accurate inference for MoE-based LLMs.

Table 1: Underlying Protocol and Description

| Protocol | Description | Protocol | Description |
|---|---|---|---|
| $\Pi_{\text{mux}}$ | $[\![z]\!] = \Pi_{\text{mux}}([\![b]\!]^B, [\![x]\!])$, s.t. $z = b \cdot x$ | $\Pi_{\text{mul}}$ | $[\![z]\!] = \Pi_{\text{mul}}([\![x]\!], [\![y]\!])$, s.t. $z = x \cdot y$ |
| $\Pi_{\text{equal}}$ | $[\![z]\!]^B = \Pi_{\text{equal}}([\![x]\!], [\![y]\!])$, s.t. $z = \mathbf{1}\{x == y\}$ | $\Pi_{\text{softmax}}$ | $[\![z]\!] = \Pi_{\text{softmax}}([\![x]\!])$, s.t. $z = \text{softmax}(x)$ |
| $\Pi_{\text{topk}}$ | $[\![W]\!], [\![K]\!] = \Pi_{\text{topk}}([\![x]\!], k)$, s.t. $W, K = \text{Top-K}(x, k)$ | $\Pi_{\text{matmul}}$ | $[\![Z]\!] = \Pi_{\text{matmul}}([\![X]\!], [\![Y]\!])$, s.t. $Z = XY$ |
| $\Pi_{\text{onehot}}$ | $[\![z]\!] = \Pi_{\text{onehot}}([\![x]\!], c)$, s.t. $z = \text{onehot}(x, c)$, where $z[i][j] = \mathbf{1}\{x[i] == j\}, \forall j \in [0, c-1]$ | | |

# 2 Preliminaries

**Notations.** We use $\{x_i\}_{i=0}^{n-1}$ to denote a set $\{x_0, x_1, \cdots, x_{n-1}\}$. We use $n, m, k$ to **denote the number of experts, tokens, and the number of experts to be activated**, respectively. We use $\mathbf{1}\{\mathcal{P}\}$ to denote the indicator function, which is 1 when $\mathcal{P}$ is true and 0 otherwise.

## 2.1 Mixture of Experts Layer

We present a brief introduction to the Mixture of Experts layer. The output of the MoE module for a given input $x$ is determined by the weighted sum of the outputs of selected expert networks. The gate routing determines the weights and the selected experts:

$$W, K = \text{Top-K}(G(x), k), \tag{1}$$

where $k$ is the number of experts to activate, $G$ is the gating network implemented by the softmax over a linear layer, i.e., $G(x) = \text{Softmax}(\text{Linear}(x))$, $K$ denotes the indices of selected experts and $W = \{G(x)_i\}_{i \in K}$ is the of the selected experts. The output of a MoE layer is then given by:

$$\text{MoE}(x) := \sum_{i \in K} W_i \cdot E_i(x), \quad E_i(x) := \text{SwiGLU}_i(x) \tag{2}$$

Each expert network $E_i$ is a feed-forward network (FFN) implemented by SwiGLU [23]. For input with multiple tokens $\{x_i\}_{i=0}^{m-1}$, tokens are routed to different experts based on the gating network. Then each expert network $E_i$ processes the tokens distributed to it in parallel.

## 2.2 Cryptographic Primitives

**Homomorphic Encryption (HE).** Following most hybrid HE/MPC schemes [24, 25, 26, 27, 28], CryptoMoE leverages the Brakerski-Fan-Vercauteren (BFV) HE scheme [29] and mainly involves the following element-wise HE operations: ciphertext addition, ciphertext-plaintext multiplication, and ciphertext rotation $\text{Rot}(\text{ct}, s)$, which shifts the ciphertext ct to the left by $s$ positions.

**Secure Multi-Party Computation (MPC).** We employ a 2-out-of-2 additive Secret Share (SS)-based MPC scheme [30] to keep the input data private throughout inference. We denote two parties by $P_0$ and $P_1$, where $P_0$ is the client and $P_1$ is the server. We use $[\![x]\!]$ to denote an additive share of $x$. We write $[\![x]\!] = ([\![x]\!]_0, [\![x]\!]_1)$ where $P_0$ holds $[\![x]\!]_0$ and $P_1$ holds $[\![x]\!]_1$, such that $[\![x]\!]_0 + [\![x]\!]_1 = x$. We write $[\![x]\!]^B$ to denote the share of Boolean data. This work builds upon pre-existing MPC protocols whose input and output are additive shares [30, 24, 31]. These protocols are summarized in Table 1. Among them, $\Pi_{\text{matmul}}$ is implemented using HE [24], while the rest are implemented using oblivious transfer (OT) [30, 31].

**Threat Model and Security Guarantee.** CryptoMoE works in a general private inference scenario that involves two parties, i.e., server $P_1$ and client $P_0$. The server holds the proprietary NN model, and the client owns private input [26, 28, 27, 32, 24, 31]. CryptoMoE enables the client to obtain the inference results while keeping the server's model weights and the client's input private. Consistent with previous works [26, 30, 33, 27, 24, 31], CryptoMoE adopts an *honest-but-curious* security model in which both parties follow the specification of the protocol but also try to learn more than allowed. CryptoMoE is built upon cryptographic primitives, including BFV and MPC protocols, the security can hence be guaranteed following [29, 34].

## 2.3 Related Work

With the proliferation of ChatGPT, significant efforts have been made to enable private Transformer inference, including hybrid HE/MPC frameworks [32, 24, 31, 35, 36, 37, 38], Fully-HE frameworks [39, 40, 41] and Fully-MPC frameworks [42, 43, 44, 45]. However, these works only support

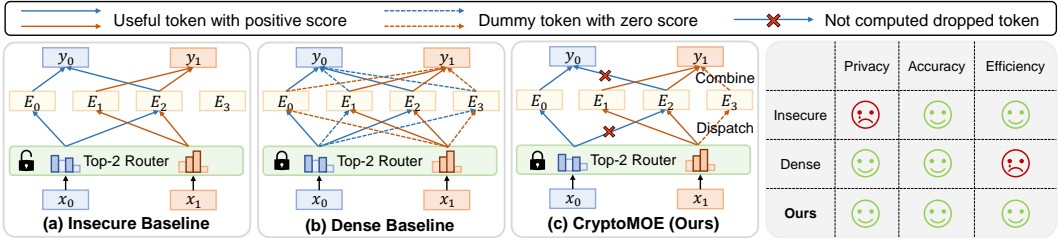

Figure 2: Toy examples and qualitative comparisons between baselines and CryptoMoE.

dense models like GPT-2 [6]. Recent work, CipherPrune [22], introduces dynamic token pruning for private Transformer inference. Nevertheless, applying its pruning protocol directly to MoE layers not only leaks the number of tokens assigned to each expert but also incurs substantial communication overhead. Thus, how to support private MoE-based model inference is still an open question.

## 3 Private MoE Inference and Baselines

We first establish a general conceptual framework for private inference in MoE models, where all intermediate results are kept in secret shared form. The framework consists of four steps:

❶ **Gate Routing.** Given $m$ input tokens $\{[\![x_i]\!]\}_{i=0}^{m-1}$, generating a routing score $[\![W]\!]$ and indices of selected experts $[\![K]\!]$, as in Equation 1. ❷ **Dispatch** $\Pi_{\text{dispatch}}$. Given $[\![W]\!], [\![K]\!]$, $\Pi_{\text{dispatch}}$ is expected to securely determine the set of tokens assigned to each expert, denoted as $\{[\![\mathcal{X}_i]\!]\}_{i=0}^{n-1}$, where $[\![\mathcal{X}_i]\!] = \{[\![x_j]\!] \mid x_j \text{ is routed to expert } E_i\}$. ❸ **Expert Compute.** Given $\{[\![\mathcal{X}_i]\!]\}_{i=0}^{n-1}$, each of the $n$ experts performs computation as $[\![y_{E_i}]\!] = E_i([\![\mathcal{X}_i]\!])$. ❹ **Combine** $\Pi_{\text{combine}}$. After obtaining all expert outputs $\{[\![y_{E_i}]\!]\}_{i=0}^{n-1}$, a $\Pi_{\text{combine}}$ protocol is intended to securely aggregate the outputs and produce the final token-wise results $\{[\![y_i]\!]\}_{i=0}^{m-1}$.

As mentioned in Figure 1(b), even revealing the number of tokens assigned to each expert in step ❷ may leak information about the types of input and the experts. Therefore, **ensuring private inference for MoE models without disclosing any information about routing information** $W, K$ **constitutes a core challenge**. We first construct two baselines that serve as benchmarks across three key dimensions: privacy, efficiency, and accuracy, as shown in Figure 2.

**Insecure Baseline** where $[\![W]\!], [\![K]\!]$ is revealed in public, then the dispatch step can be executed in plaintext without extra cost. Expert computation remains encrypted. This baseline achieves the highest accuracy and efficiency, as its computational flow is identical to the plaintext counterpart and avoids extra overhead from $\Pi_{\text{dispatch}}$. However, it leaks complete routing information and thus serves only as an upper bound reference for accuracy and efficiency.

**Dense baseline.** To protect routing information, an approach is to follow the non-MoE models by evaluating all experts for every token, regardless of routing decisions, as depicted in Figure 2 (b). This removes the need for $\Pi_{\text{dispatch}}$, and the final output is a weighted sum based on routing scores, with non-selected experts receiving zero weight. While this method protects routing privacy and maintains the same accuracy as the insecure baseline, it drastically increases computation. For example, in QWen-MoE with 4-out-of-60 expert selection, it incurs a $15\times$ increase in expert computation. Therefore, achieving private, efficient, accurate inference for MoE models remains an open question.

## 4 CryptoMoE Framework

### 4.1 Inference-Time Balanced Expert Routing

In this section, we introduce CryptoMoE, a private, efficient, and accurate MoE inference framework. Building upon the dense baseline, CryptoMoE advances a key idea: **Inference-Time Balanced Expert Routing**, where each expert processes exactly $t$ tokens, regardless of the routing outcome. Figure 3 shows the private inference workflow of a MoE layer with CryptoMoE. Step ❶❸❹ follows the procedure described in Section 3. The key differences is step ❷, which invokes a $\Pi_{\text{dispatch}}$ to

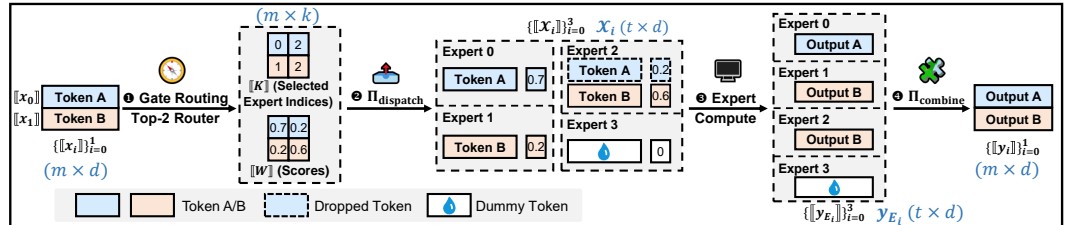

Figure 3: The workflow of private MoE layer inference in CryptoMoE.

produce $\{\mathcal{X}_i\}_{i=0}^{n-1}$ without leaking routing information. Notably, $\Pi_{\text{dispatch}}$ enforces that each expert's input $\mathcal{X}_i$ contains exactly $t$ tokens. If fewer than $t$ tokens are routed to an expert, dummy tokens are added for padding; if more than $t$ are routed, excess tokens are dropped and not computed. By appropriately choosing $t$, for example by setting $t = mk/n$, the expected number of tokens per expert, we can maintain routing privacy without increasing computational cost. However, realizing such balanced expert routing in the private inference setting introduces three key challenges:

**Challenge 1: Significant accuracy degradation.** We observe that dropping tokens beyond the threshold $t$ during dispatch can lead to up to 7% accuracy loss, as some discarded tokens are critical to the final output. Thus, minimizing the accuracy loss caused by token dropping is the first challenge.

**Challenge 2: Construction of $\Pi_{\text{dispatch}}$ and $\Pi_{\text{combine}}$.** While step ❷❸ can be implemented by existing protocol $\Pi_{\text{Softmax}}$, $\Pi_{\text{MatMul}}$ and $\Pi_{\text{Top-K}}$ proposed in Bolt [24] and Bumblebee [31], constructing $\Pi_{\text{dispatch}}$ and $\Pi_{\text{combine}}$ is non-trivial. $\Pi_{\text{dispatch}}$ must securely assign tokens to experts based on routing information and select the $t$ tokens for each expert. Similarly, constructing the $\Pi_{\text{combine}}$ protocol to aggregate expert outputs into token-wise results is complex. Designing both protocols using MPC and HE protocols to preserve privacy without incurring significant overhead remains a challenge.

**Challenge 3: High cost of linear layer computations.** In expert computation, the three linear layers in SwiGLU dominate the overall latency. Since each expert receives only a few tokens, the number of token dimensions that can be packed per ciphertext is limited, leading to an excessive number of costly HE rotations. Reducing this overhead is critical for improving efficiency.

To tackle these challenges, we propose: (i) a confidence-aware secure dispatch protocol that protects routing information while alleviating accuracy loss (Section 4.2); (ii) a lightweight and secure combine protocol (Section 4.3); and (iii) a batch matrix multiplication protocol that reduces HE rotations by a factor of $n$, significantly accelerating expert computation (Section 4.4).

### 4.2 Confidence-Aware Secure Dispatch Protocol

Token discarding occurs when more than $t$ tokens are routed to the same expert. Uniformly selecting $t$ tokens with equal probability can lead to up to 7% accuracy loss. To mitigate this, we propose a confidence-aware selection strategy: re-rank the tokens assigned to each expert based on their routing confidence $W = G(x)$ and retain the top-$t$ tokens. As shown in Figure 3, Expert 2 receives two tokens and selects token B, which has a higher confidence score. As demonstrated in Section 5, this approach consistently improves accuracy across different models and datasets.

Next, we construct our secure $\Pi_{\text{dispatch}}$. In private inference, dispatching the appropriate $t$ tokens to each expert is challenging, as we must keep the routing information $W, K$ as secret shares. Figure 4 illustrates our confidence-aware secure dispatch protocol. Specifically, $\Pi_{\text{dispatch}}$ takes $m$ secret-shared tokens $\{[\![x]\!]_i\}_{i=0}^{m-1}$ and routing information $\{[\![W]\!], [\![K]\!]\}$ as inputs, and outputs the set of tokens assigned to each expert, $\{[\![\mathcal{X}_i]\!]\}_{i=0}^{n-1}$, where each expert receives exactly $t$ tokens. Our core idea is that after Top-$k$ routing, there are $km$ candidate tokens to be assigned to $n$ experts. For each expert, we rank the $km$ tokens by their confidence scores and select the top $t$ tokens. Tokens not assigned to the expert have zero scores. This design ensures each expert receives the desired $t$ tokens.

For each expert $E_i$, the protocol contains three steps: ❶ **Compute token priority scores.** Use $\Pi_{\text{equal}}$ to evaluate whether selected expert indices $[\![K]\!]$ are equal to $i$, producing boolean mask $[\![M_i]\!] \in \{0, 1\}^{km}$ that indicate which tokens are useful for this expert. Next, we employ $\Pi_{\text{mux}}$ to combine the routing scores $[\![W]\!]$ with the masks $[\![M_i]\!]$, resulting in token priority scores $[\![S_i]\!]$.

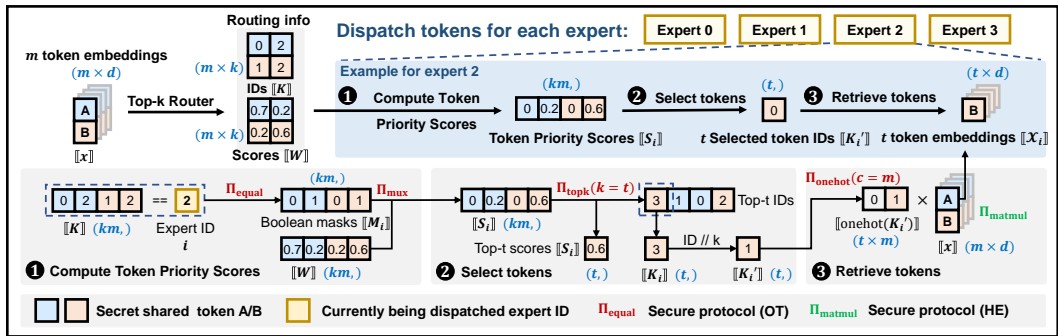

Figure 4: Secure dispatch protocol for $n = 4, m = 2, t = 1$.

Useful tokens retain their original scores, while useless (dummy) tokens' scores are set to zero. To proceed, we must select the top $t$ tokens from the $km$ candidates based on these scores. Prior work CipherPrune [22], introduces a secure pruning protocol capable of achieving this. However, it incurs a high communication cost of $O(kmtd)$, where $d$ denotes the hidden dimension. This overhead arises because each comparison involves a secure swap of the whole token embeddings, making it impractical for MoE inference. To address this limitation, we propose a novel protocol that decouples token scores from embeddings, thereby reducing the communication complexity to $O(km \log(km))$. A detailed comparison with CipherPrune is provided in Appendix C. Our protocol comprises two steps: ❷ **Select token indices.** We apply $\Pi_{\text{topk}}(k = t)$ to select the $t$ tokens with the highest priority scores. The resulting indices $[\![K_i]\!] \in \mathbb{Z}^t$ correspond to token positions within the dispatched sequence of length $km$. To map these indices back to the original $m$-token input sequence, we perform an integer division by $k$ to get $[\![K_i']\!] = [\![\lfloor \frac{K_i}{k} \rfloor]\!]$. ❸ **Retrieve tokens.** Using the selected indices, we retrieve the corresponding token embeddings. Specifically, we convert the selected indices $[\![K_i']\!] \in \mathbb{Z}^t$ into a one-hot matrix using $\Pi_{\text{onehot}}([\![K_i']\!], m) \in \mathbb{Z}^{t \times m}$, which requires $O(tm)$ calls to $\Pi_{\text{equal}}$. We then perform a $\Pi_{\text{MatMul}}$ with the input token embeddings $[\![x]\!] \in \mathbb{Z}^{m \times d}$ to obtain the $t$ desired token embeddings for expert $i$, i.e., $[\![\mathcal{X}_i]\!] \in \mathbb{Z}^{t \times d}$.

## 4.3 Efficient Secure Combine Protocol

After the computation by each expert, we obtain the output $[\![y_{E_i}]\!]$ for expert $i$. A combination process is necessary to aggregate results across all experts into the final token-wise outputs $[\![y_i]\!]$. This process must address the token reordering challenge: each $[\![y_{E_i}]\!]$ contains $t$ tokens, ordered according to $\Pi_{\text{dispatch}}$, which differs from the original sequence order.

To address this, we propose a lightweight **one-hot-based reordering** method, illustrated in Figure 5 (a). We reuse the one-hot matrix $[\![\text{onehot}(K_i')]\!] \in \mathbb{Z}^{t \times m}$, computed in step ❸ of $\Pi_{\text{dispatch}}$, and perform a local transpose to obtain $[\![\text{onehot}(K_i')^T]\!] \in \mathbb{Z}^{m \times t}$. Next, we perform a $\Pi_{\text{MatMul}}$ with $[\![y_{E_i}]\!] \in \mathbb{Z}^{t \times d}$ to reorder the tokens and compute the final token-wise result $[\![y_i]\!] \in \mathbb{Z}^{m \times d}$. The complete combine protocol is illustrated in Figure 5 (b). Before reordering, we use $\Pi_{\text{mul}}$ to multiply the one-hot matrix with token scores $[\![S_i]\!]$, producing a scored one-hot matrix $[\![R_i]\!] \in \mathbb{Z}^{m \times t}$. Subsequently, token reordering and weighted masking are performed simultaneously using a $\Pi_{\text{matmul}}$ on $[\![R_i]\!]$ and $[\![y_{E_i}]\!]$. Finally, the outputs from all experts are summed to obtain the final result for the MoE layer. With this construction, $\Pi_{\text{combine}}$ requires only one $\Pi_{\text{mul}}$ and one $\Pi_{\text{matmul}}$, making it highly efficient.

**Complexity Analysis.** For a single MoE layer, the proposed $\Pi_{\text{dispatch}}$ and $\Pi_{\text{combine}}$ introduce additional communication overhead of $O(nkm \log(km) + ntm)$, where the first term stems from the $\Pi_{\text{topk}}$ protocol [46] and the second from $\Pi_{\text{onehot}}$. Experimental results show that our protocol is highly efficient, incurring only an 18% overhead while preserving privacy and leveraging the sparsity of MoE computation. Further implementation details of $\Pi_{\text{dispatch}}$ and $\Pi_{\text{combine}}$ are provided in Appendix B.

## 4.4 Efficient Batch Ciphertext-Plaintext MatMul (Batch MatMul) Protocol

Linear layer evaluation remains a major bottleneck in HE for expert computation. Unlike dense models, MoE layers process $n$ groups of parallel tokens, each of size $t \times d_1$ ($t$ tokens, each with

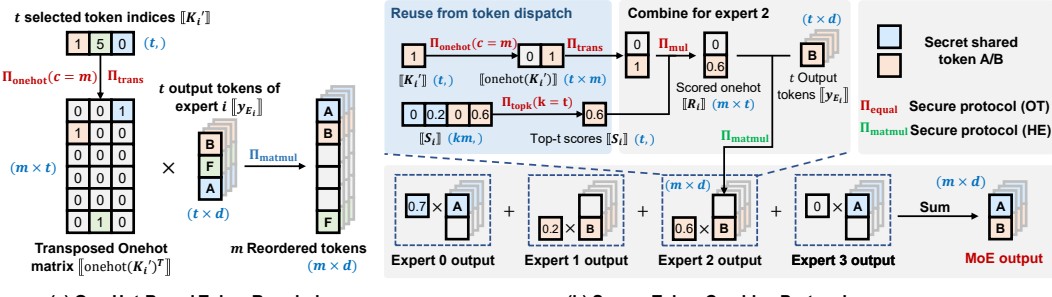

(a) One-Hot-Based Token Reordering

(b) Secure Token Combine Protocol

Figure 5: (a) An example for one-hot-based token reordering. In this case, $m = 6, t = 3$. (b) Secure combine protocol for $n = 4, m = 2, t = 1$.

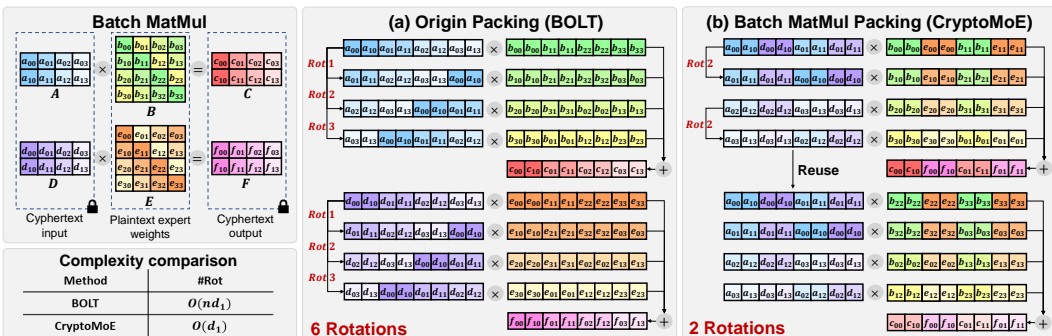

Figure 6: Batch MatMul protocol reduces the number of HE rotations from $O(nd_1)$ to $O(d_1)$. This image illustrates an example of MatMul with a batch size $n = 2$. We need to compute $C = A \times B$ and $F = D \times E$, where $A, D \in \mathbb{Z}_p^{t \times d_1}$ are ciphertext inputs, $B, E \in \mathbb{Z}_p^{d_1 \times d_2}$ are plaintext expert weights. In this example, $t = 2, d_1 = 4, d_2 = 4$, and each ciphertext can pack 8 elements. Through batch MatMul packing in subfigure (b), we reduce the number of rotations from 6 to 2.

an embedding dimension of $d_1$. Existing packing schemes like BOLT [24] optimize for dense models by packing along the $t$-dimension, reducing the packed hidden size in a ciphertext and thus minimizing expensive ciphertext rotations. However, in MoE layers, each expert handles only a few tokens. Applying these schemes increases the packed hidden dimension, leading to more rotations and an extremely higher computation cost.

To address this, we introduce an efficient batch ciphertext-plaintext MatMul protocol tailored for MoE computation. The key idea is to pack partial token embeddings from all experts into a single ciphertext. Figure 6 shows a toy example where $n = 2, t = 2, d_1 = 4$, with each ciphertext holding 8 elements. In the original packing scheme, each expert's input matrix of size $t \times d_1$ is packed into a ciphertext, requiring 3 rotations per MatMul to accumulate partial sums, resulting in 6 total rotations for two experts. In contrast, our method packs partial embeddings of all tokens into a single ciphertext with shape $(nt \times \frac{d_1}{n})$. The weight matrices are adjusted accordingly in plaintext without additional overhead. As shown in Figure 6 (b), this reduces the number of rotations to just 2, thanks to the smaller hidden dimension in each ciphertext. Our batch MatMul protocol reduces HE rotations from $O(nd_1)$ to $O(d_1)$. Our method is also compatible with the Baby-Step Giant-Step (BSGS) algorithm [24], which can further reduce the number of rotation operations. The Complexity analysis is provided in Appendix D.

# 5  Experiments

## 5.1  Experimental Setup

**Implementation.** We implement CryptoMoE upon the SecretFlow-SPU framework [47], which is a popular framework for privacy-preserving deep learning. We adopt a secure two-party computation

| Model | Method | Latency (s/token) | | Comm. (MB/token) |
|---|---|---|---|---|
| | | LAN | WAN | |
| DeepSeekMoE 2.8B/16.4B | Insecure* | 1.22 | 6.55 | 31.3 |
| | Dense | 4.43 | 20.52 | 310.9 |
| | CryptoMoE$^\top_{t=1.0}$ | 1.89 | 11.86 | 182.0 |
| | **CryptoMoE$_{t=1.0}$** | **0.77 (5.8×)** | **8.41 (2.4×)** | **39.1 (8.0×)** |
| | **CryptoMoE$_{t=2.0}$** | **1.06 (4.1×)** | **9.40 (2.2×)** | **71.9 (4.3×)** |
| OLMoE 1.3B/6.9B | Insecure* | 0.95 | 5.46 | 29.7 |
| | Dense | 3.38 | 17.21 | 232.0 |
| | CryptoMoE$^\top_{t=1.0}$ | 1.09 | 11.03 | 93.0 |
| | **CryptoMoE$_{t=1.0}$** | **0.83 (4.1×)** | **8.55 (2.0×)** | **42.2 (5.5×)** |
| | **CryptoMoE$_{t=2.0}$** | **1.02 (3.3×)** | **9.92 (1.7×)** | **75.0 (3.1×)** |
| QWenMoE 2.7B/14.3B | Insecure* | 0.94 | 4.86 | 19.5 |
| | Dense | 4.28 | 18.74 | 291.4 |
| | CryptoMoE$^\top_{t=1.0}$ | 2.72 | 18.52 | 288.3 |
| | **CryptoMoE$_{t=1.0}$** | **0.56 (7.6×)** | **6.93 (2.7×)** | **25.8 (11.3×)** |
| | **CryptoMoE$_{t=2.0}$** | **0.73 (5.9×)** | **7.69 (2.4×)** | **47.7 (6.1×)** |

\* Insecure baseline with public routing information.
$^\top$ CryptoMoE baseline with CipherPrune's protocol.

Table 2: Cost comparison of single MoE layer. Insecure baseline cannot leverage batch MatMul optimization, and is therefore slower than CryptoMoE in some cases.

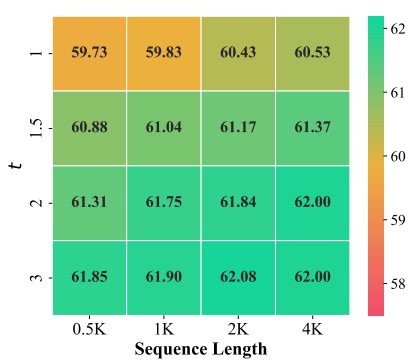

Figure 7: Effect of $t$ and sequence length on average accuracy. $62.19\%$ is the average accuracy of the original model.

(2PC) setting **without** a trusted third party. All the experiments are performed on a machine with an Intel Xeon Platinum 8468 CPU (48 cores and 2.1GHz). We consider two network environments: 1) LAN setting with 3Gbps bandwidth and 0.2ms latency; 2) WAN setting with 400Mbps bandwidth and 40ms latency. We simulate network environment via Linux Traffic Control.

**Datasets and Models.** We consider three popular MoE models: 1) DeepSeekMoE-2.8B/16.4B (6 of 64 experts) [8], 2) QWenMoE-2.7B/14.3B (4 of 60 experts) [20] and 3) OLMoE-1.3B/6.9B (6 of 60 experts) (ICLR'25 Oral) [21]. Since MoE models typically follow a similar design, CryptoMoE can also be applied to other MoE models. All the models are evaluated on eight famous zero-shot common sense reasoning tasks, including SIQA [13], OBQA [17], BoolQ [11], ARC-easy, ARC-challenge [16], HellaSwag [14], PIQA [12], and WinoGrande [15].

**Baselines.** Since CryptoMoE is the first framework enabling private MoE inference, we compare it with three baselines: 1) Insecure baseline, 2) Dense baseline, 3) CryptoMoE$^\top$ baseline where $\Pi_{\text{dispatch}}$ and $\Pi_{\text{combine}}$ are constructed by CipherPrune [22]'s pruning protocol.

**Selection of $t$.** The token count $t$ assigned to each expert plays a critical role in balancing accuracy and efficiency. A larger $t$ generally leads to higher accuracy but at the cost of reduced efficiency. We argue that the lower bound of $t$ is $mk/n$, which matches the number of tokens computed in the original MoE model without introducing additional computation cost. However, due to the inherent imbalance in token routing, this setting often results in some tokens being discarded, leading to accuracy degradation. Empirically, setting $t = 2mk/n$ achieves a favorable balance, where $mk/n$ is the expected number of tokens per expert. In subsequent experiments, we denote configurations with $t = mk/n$, $2mk/n$, etc., as CryptoMoE$_{t=1.0}$, CryptoMoE$_{t=2.0}$, and so forth.

## 5.2 Cost Comparison of Single MoE Layer

In Table 2, we compare the latency and communication costs for a single MoE layer. We evaluate the prefill stage and report the amortized per-token latency and communication by dividing the total values by the input sequence length. The results show that: **1)** With CryptoMoE$_{t=2.0}$, we observe a $1.7 \sim 5.9\times$ latency reduction and a $3.1 \sim 6.1\times$ communication reduction compared to dense baseline across three models. Additionally, CryptoMoE$_{t=1.0}$ achieves a $2 \sim 11\times$ communication reduction over CipherPrune's protocol. **2)** As $t$ increases, the latency of CryptoMoE grows slowly due to batch MatMul optimization. A larger $t$ allows more tokens to be packed together, reducing expensive HE rotations and limiting latency growth. **3)** CryptoMoE matches the insecure baseline's performance in some cases. This is because the insecure baseline cannot leverage batched MatMul, as experts receive varying token counts, making it complicated to apply batching in HE.

## 5.3 End-to-End Evaluation

We benchmark the accuracy, end-to-end amortized latency, and communication cost of different methods in Table 3, using a batch size of 16 and CryptoMoE$_{t=2.0}$. An ablation study on both batch size

Table 3: End-to-end comparison with baselines.

| Model | Method | Accuracy (%) ↑ | | | | | | | | | Latency (min/token) ↓ | | Comm. |
|---|---|---|---|---|---|---|---|---|---|---|---|---|---|
| | | SIQA | OBQA | BoolQ | ARC-easy | ARC-challenge | HellaSwag | PIQA | WinoGrande | Avg. | LAN | WAN | (GB) ↓ |
| DeepSeekMoE 2.8B/16.4B | Insecure* Dense | 32.9 | 43.6 | 72.5 | 73.0 | 47.9 | 77.2 | 80.3 | 70.1 | 62.2 | 0.83 / 2.33 | 3.48 / 10.0 | 1.33 / 9.16 |
| | $CryptoMoE^\top_{t=1.0}$ | 32.7 | 40.6 | 72.0 | 68.9 | 43.7 | 73.7 | 78.1 | 68.9 | 59.8 | 1.14 | 5.96 | 5.55 |
| | $CryptoMoE_{t=2.0}$ | 32.8 | 42.8 | 72.4 | 72.2 | 47.1 | 76.2 | 80.2 | 70.4 | 61.8 (-0.4) | **0.76** (3.1×) | **4.81** (2.1×) | **2.46** (3.7×) |
| OLMoE 1.3B/6.9B | Insecure* Dense | 32.9 | 45.0 | 74.6 | 76.1 | 48.6 | 77.0 | 81.0 | 68.6 | 63.0 | 0.34 / 0.99 | 1.62 / 4.75 | 0.58 / 3.82 |
| | $CryptoMoE^\top_{t=1.0}$ | 32.9 | 41.4 | 73.1 | 70.8 | 46.0 | 72.4 | 75.5 | 66.4 | 59.8 | 0.38 | 3.10 | 1.60 |
| | $CryptoMoE_{t=2.0}$ | 32.9 | 45.6 | 74.7 | 75.2 | 47.4 | 75.8 | 79.4 | 68.6 | 62.5 (-0.5) | **0.36** (2.8×) | **2.81** (1.7×) | **1.31** (2.9×) |
| QWenMoE 2.7B/14.3B | Insecure* Dense | 32.3 | 43.8 | 79.8 | 68.9 | 44.2 | 77.3 | 80.4 | 69.2 | 62.0 | 0.64 / 1.98 | 2.41 / 7.96 | 1.09 / 7.61 |
| | $CryptoMoE^\top_{t=1.0}$ | 33.8 | 40.8 | 79.0 | 64.5 | 42.8 | 75.0 | 78.2 | 60.2 | 60.2 | 1.36 | 7.88 | 7.54 |
| | $CryptoMoE_{t=2.0}$ | 33.8 | 42.6 | 79.6 | 69.1 | 44.0 | 76.7 | 80.7 | 69.5 | 62.0 (-0.0) | **0.56** (3.5×) | **3.54** (2.2×) | **1.76** (4.3×) |

\* Insecure baseline with public routing information. $^\top$ CryptoMoE baseline with CipherPrune's protocol for $\Pi_{dispatch}$ and $\Pi_{combine}$.

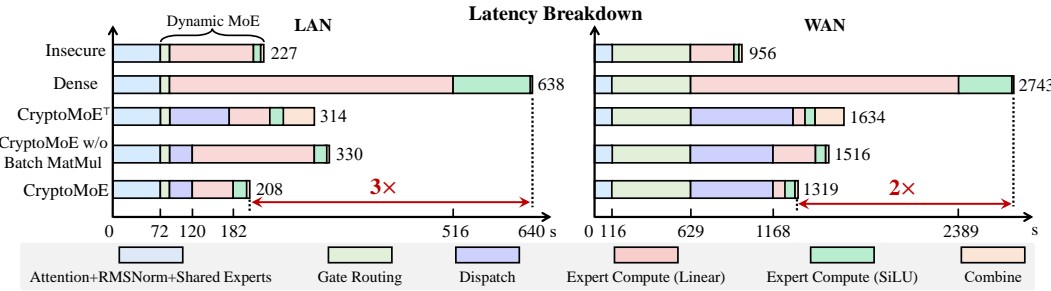

Figure 8: Latency breakdown under LAN and WAN settings.

and $t$ will be presented in Section 5.4. The results demonstrate the following: **1)** CryptoMoE retains 99.2% of the accuracy of the insecure baseline on average. **2)** With comparable accuracy, CryptoMoE reduces LAN latency by $2.8 \sim 3.5\times$, WAN latency by $1.7 \sim 2.2\times$, and communication cost by $2.9 \sim 4.3\times$ compared to the dense baseline. Moreover, it offers up to $2.4\times$ latency reduction over CipherPrune's protocol with higher accuracy thanks to our efficient $\Pi_{dispatch}$ and $\Pi_{combine}$ protocols.

## 5.4 Ablation Study

**Ablation Study on $t$ and Sequence Length.** CryptoMoE benefits from balanced expert loads, as fewer tokens are discarded. Since both $t$ and the input sequence length influence accuracy, we perform a two-dimensional ablation study on DeepSeekMoE; results for other models are provided in Appendix E. We vary the average sequence lengths from 0.5K to 4K by changing the batch size from 8 to 64. For each sequence length, we adopt different $t$ values and report the average accuracy across all datasets in Figure 7. We observe that increasing either $t$ or the sequence length improves accuracy, but gains become marginal beyond 2K tokens. This is likely due to inherent expert load imbalance in the dataset rather than input length limitations. Overall, setting $t = 2$ provides a robust trade-off across different configurations.

**Ablation Study on Different Components.** We demonstrate the effectiveness of the proposed techniques by adding them step by step. As shown in Table 4, we observe that: **1)** Without confidence-aware selection, balanced expert routing reduces latency but harms accuracy a lot. **2)** The batched MatMul optimization substantially reduces the computational overhead of expert linear layers, leading to a $2\times$ reduction in end-to-end latency.

**Latency Breakdown.** To analyze the bottleneck of CryptoMoE and other baselines, we profiled a single Transformer block from DeepSeekMoE under LAN and WAN settings. The breakdown is shown in Figure 8. Except for the first item, "Attention+RMSNorm+Shared Experts," all other components are

| Model | Method | Accuracy (%) | Latency (min/token) |
|---|---|---|---|
| DeepSeekMoE 2.8B/16.4B | Dense Baseline | 62.2 | 2.33 |
| | +Balanced Expert Routing | 57.9 | 1.20 |
| | +Confidence-aware selection | 61.8 | 1.20 |
| | +Batch MatMul | 61.8 | 0.76 |
| OLMoE 1.3B/6.9B | Dense Baseline | 63.0 | 0.99 |
| | +Balanced Expert Routing | 50.9 | 0.55 |
| | +Confidence-aware selection | 62.5 | 0.55 |
| | +Batch MatMul | 62.5 | 0.36 |
| QWenMoE 2.7B/14.3B | Dense Baseline | 62.0 | 1.98 |
| | +Balanced Expert Routing | 55.1 | 1.23 |
| | +Confidence-aware selection | 62.0 | 1.23 |
| | +Batch MatMul | 62.0 | 0.56 |

Table 4: Ablation study of accuracy and amortized latency (LAN) on different components.

related to the dynamic MoE layer. We draw the following conclusions: **1)** In our scenario with short sequence lengths, the MoE layer dominates runtime, accounting for 68% and 91% of the total latency under LAN and WAN settings, respectively. This highlights the necessity for MoE layer optimization. **2)** Within the MoE layer, the expert linear layers are the primary bottleneck. Our batch MatMul optimization reduces their cost by $3 \sim 6\times$, yielding a $2 \sim 3\times$ reduction in overall latency. **3)** The dispatch and combine protocols contribute only 18% of LAN latency while ensuring routing privacy **4)** Under WAN, gate routing and dispatch latency increases significantly, mainly due to the top-$k$ protocol, which involves many communication rounds. Developing round-efficient top-$k$ protocols remains a key direction for future improvement.

**Scalability.** Our balanced expert routing strategy is scalable to larger models. Figure 9 shows the average accuracy of naive selection strategy (i.e., uniform random selection) and our CryptoMoE on Mixtral-13B/47B [48] and LLaMA4-Scout-17B/109B [1]. It can be seen that CryptoMoE consistently outperforms naive selection across all configurations. CryptoMoE maintains 100% accuracy on Mixtral even with $t = 1.0$. On LLaMA4-Scout-109B, our CryptoMoE$_{t=2.0}$ maintains 98.8% accuracy of the original model.

For private inference, due to the large model size, the memory usage of the SPU during execution exceeds the physical memory capacity of our machine. Reducing the memory overhead of private inference, especially for larger models, remains a challenging problem.

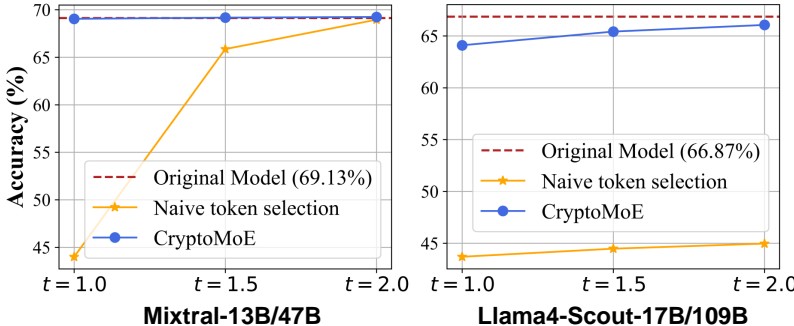

Figure 9: Accuracy of random selection and our CryptoMoE on Mixtral-13B/47B and LLaMA4-Scout-17B/109B.

# 6 Limitation and Future Work

Under WAN settings, $\Pi_{\text{topk}}$ becomes a bottleneck due to its massive communication rounds, which could be optimized in future work. Additionally, inference-time balanced expert routing is less effective for very short input sequences (e.g., length < 64), as it leads to severe imbalance, which is another promising direction for improvement.

# 7 Conclusion

We propose CryptoMoE, the first framework to enable private, accurate, and efficient inference for MoE-based LLMs. CryptoMoE preserves privacy through balanced expert routing and introduces novel secure dispatch and combine protocols tailored for MoE layers. It also incorporates a batch Mat-Mul protocol to boost computational efficiency. Experimental results show that CryptoMoE achieves $2.8 \sim 3.5\times$ reduction in end-to-end latency compared to the dense baseline and an up to $4.3\times$ reduction in communication cost over CipherPrune, all with negligible accuracy loss.

# Acknowledgements

This work was supported in part by NSFC under Grant 62495102, Grant 92464104, and Grant 62341407, in part by the National Key Research and Development Program under Grant 2024YFB4505004, in part by Beijing Municipal Science and Technology Program under Grant Z241100004224015, and in part by 111 Project under Grant B18001.

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

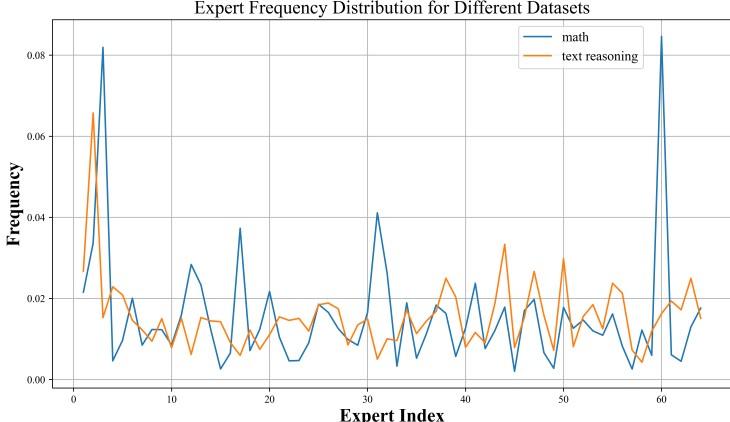

Figure 10: Expert frequency distribution of the 10th layer of DeepSeekMoE-16B for text understanding tasks and mathematical reasoning tasks.

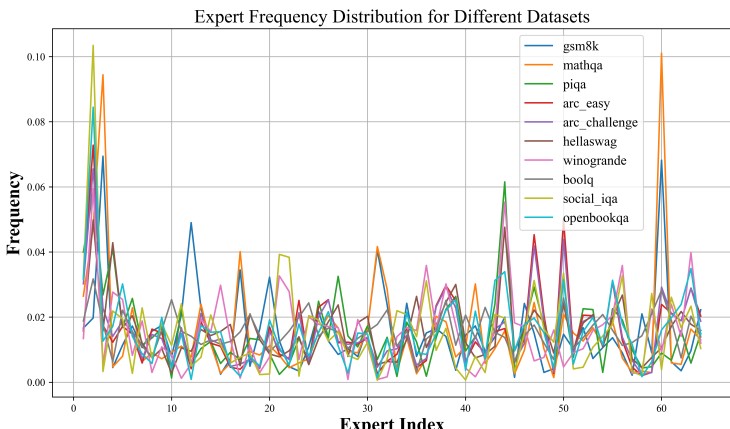

Figure 11: Expert frequency distribution of the 10th layer of DeepSeekMoE-16B for 10 different datasets.

## A  Expert activation patterns

Figure 10 shows the expert activation patterns in the 10th layer of DeepSeekMoE-16B [8] when processing text understanding versus mathematical reasoning tasks. Text understanding tasks include SIQA [13], OBQA [17], BoolQ [11], ARC-easy, ARC-challenge [16], HellaSwag [14], PIQA [12] and WinoGrande [15]. Mathematical reasoning tasks comprise GSM8K [9] and MathQA [10]. Expert #2 is disproportionately activated for text understanding, while Experts #3 and #60 show higher activation for mathematical reasoning.

The detailed expert activation patterns of these datasets are shown in Figure 11.

## B  Secure Dispatch and Combination Protocols

We detail the secure token dispatch protocol $\Pi_{dispatch}$ in Algorithm 1 and secure combine protocol $\Pi_{combine}$ in Algorithm 2 in this section.

## C  Detailed Comparison with CipherPrune

CipherPrune proposes a secure pruning protocol that uses oblivious swaps to iteratively move pruned tokens to the end of the sequence, leveraging OT-based secure comparisons and binding masks to

---

**Algorithm 1:** Secure Dispatch Protocol $\Pi_{\text{dispatch}}$

---

**Input:** $P_0, P_1$ hold secret shares of input token embeddings $[\![x]\!] \in \mathbb{Z}^{m \times d}$ and routing information (including routing expert indices and scores $[\![K]\!], [\![W]\!] \in \mathbb{Z}^{m \times k}$), where $m$ is the number of input tokens, $d$ is the hidden dimension, and $k$ is the number of activated experts per token.

**Output:** $P_0, P_1$ learn the secret shares of dispatched token embeddings for each expert $[\![\mathcal{X}_i]\!]_{i=0}^{n-1} \in \mathbb{Z}^{t \times d}$, where $t$ is the number of dispatched tokens for each expert, and $n$ is the number of experts.

1: **for** each expert $i$ in $[0, n-1]$ **do**
2:      Flatten $[\![K]\!], [\![W]\!]$ into the shape $\mathbb{Z}^{km}$
3:      Invoke $\Pi_{\text{equal}}$ with input $[\![K]\!]$ and $i$, and set output as boolean mask $[\![M_i]\!]^B \in \mathbb{Z}^{km}$, where $M_i[j] = \mathbf{1}\{K[j] == i\}, \forall j \in [0, km-1]$.
4:      Invoke $\Pi_{\text{mux}}$ with input $[\![M_i]\!]^B$ and $[\![W]\!]$, and learn the token priority scores $[\![S_i]\!]$, where $S_i[j] = M_i[j] \cdot W[j], \forall j \in [0, km-1]$.
5:      Invoke $\Pi_{\text{topk}}([\![S_i]\!], k = t)$ to obtain selected token indices and scores $[\![K_i]\!], [\![S_i']\!] \in \mathbb{Z}^t$.
6:      Convert $[\![K_i]\!]$ to original token indices via integer division by $k$: $[\![K_i']\!] \leftarrow [\![K_i]\!]//k$.
7:      Invoke $\Pi_{\text{onehot}}([\![K_i']\!], c = m)$ to obtain one-hot matrix $[\![\text{onehot}(K_i')]\!]^B \in \mathbb{Z}^{t \times m}$. where $\text{onehot}(K_i')[j][k] = \mathbf{1}\{K_i'[j] == k\}, \forall j \in [0, t-1], k \in [0, m-1]$.
8:      Compute $[\![\mathcal{X}_i]\!] \in \mathbb{Z}^{t \times d} \leftarrow [\![\text{onehot}(K_i')]\!] \times [\![x]\!]$ using HE protocol $\Pi_{\text{matmul}}$ to retrieve $t$ dispatched token embeddings for expert $i$.
9: **end for**

---

**Algorithm 2:** Secure Combine Protocol$\Pi_{\text{combine}}$

---

**Input:** $P_0, P_1$ hold secret shares of each expert's output $[\![y_{E_i}]\!]_{i=0}^{n-1} \in \mathbb{Z}^{t \times d}$, selected top-t token scores for each expert $[\![S_i']\!]_{i=0}^{n-1} \in \mathbb{Z}^t$ and the computed one-hot matrix for selected token ID for each expert $[\![\text{onehot}(K_i')]\!]_{i=0}^{n-1} \in \mathbb{Z}^{t \times m}$, where $m$ is the number of input tokens, $d$ is the hidden dimension, and $k$ is the number of activated experts per token.

**Output:** $P_0, P_1$ learn the secret shares of MoE layer output $[\![y]\!] \in \mathbb{Z}^{m \times d}$.

1: **for** each expert $i$ in $[0, n-1]$ **do**
2:      Use $\Pi_{\text{trans}}([\![\text{onehot}(K_i')]\!])$ to get onehot matrix $[\![\text{onehot}(K_i')]\!]^T \in \mathbb{Z}^{m \times t}$
3:      Invoke $\Pi_{\text{mul}}$ with input $[\![\text{onehot}(K_i')]\!]^T$ and $[\![S_i']\!]$ to compute scored onehot matrix $[\![R_i]\!] \in \mathbb{Z}^{m \times t}$, where $R_i[j][k] = \text{onehot}(K_i')[j][k] \cdot S_i'[k], \forall j \in [0, m-1], k \in [0, t-1]$.
4:      Compute $[\![y_{E_i}']\!] \in \mathbb{Z}^{m \times d} \leftarrow [\![R_i]\!] \times [\![y_{E_i}]\!]$ using HE protocol $\Pi_{\text{matmul}}$ to compute reorderd scored tokens for expert $i$.
5: **end for**
6: Sum over expert contributions to the final MoE layer output $[\![y]\!] \leftarrow \sum_{i=0}^{n-1} [\![y_{E_i}']\!]$

---

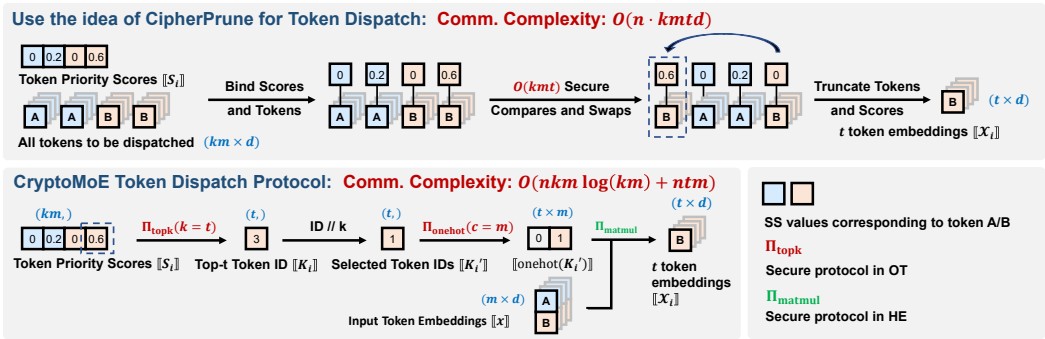

Figure 12: Comparison between CipherPrune and Our Protocol. This figure is an example for $n = 4, m = 2, t = 1$. We modified CipherPrune's mask binding strategy into score binding strategy to accommodate the requirements of MoE inference.

tokens for efficiency. Although this approach achieves secure pruning with linear complexity in the

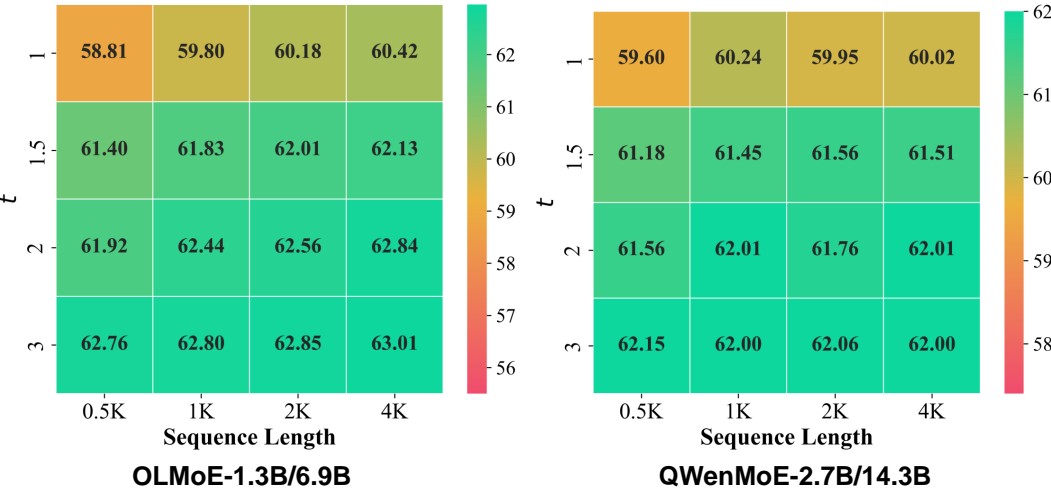

Figure 13: Effect of $t$ and sequence length on average accuracy for OLMoE and QWenMoE. The original average accuracy is 62.96% for OLMoE and 62.00% for QWenMoE.

number of swaps ($O(mp)$ for $p$ pruned tokens among $m$ candidates), applying it directly to MoE token dispatch incurs significant overhead. Specifically, for each expert, dispatching $t$ tokens to each of $k$ experts from $km$ candidate tokens requires $O(kmt)$ secure swaps on $d$-dimensional embeddings, resulting in $O(nkmtd)$ communication. Figure 12 shows a toy example of this process. Reordering in token combination after expert computation further doubles this cost. Experiments reveal that naively adopting CipherPrune introduces 82% latency overhead in privacy-preserving MoE inference.

In contrast, our protocol decouples token index selection from token embedding manipulation, eliminating expensive secure swaps on large dimensions. For token dispatch, protocol $\Pi_{\text{dispatch}}$ computes $\Pi_{\text{topk}}$ to select $t$ tokens per expert and $\Pi_{\text{onehot}}$ to encode selection masks, with complexity $O(nkmlog(km) + ntm)$. Token embeddings are then aggregated via HE-based matrix multiplication $\Pi_{\text{matmul}}$, avoiding $d$-dimensional swaps. $\Pi_{\text{matmul}}$ incur limited communication in HE-SS conversions, and the majority of computation is done by parallelizable HE operations.

For token combination, protocol $\Pi_{\text{combine}}$ reuses the selection masks from $\Pi_{\text{dispatch}}$ to invert the dispatch process via another $\Pi_{\text{matmul}}$, achieving reordering without extra secure comparisons or swaps. This reduces the latency of token combination to 1% of the total runtime.

Experiments demonstrate that our protocol introduces only 18% overhead, a $4.7\times$ improvement over CipherPrune, mainly due to replacing secure swaps with efficient HE-based linear operations. This design proves particularly advantageous for privacy-preserving MoE model inference, where large values of $d$ and $t$ make communication efficiency critical.

## D Complexity Analysis of Batch MatMul Packing

Given a sequence of ciphertext inputs $\{A_i\}_{i=0}^{n-1} \in \mathbb{Z}^{t \times d_1}$ and plaintext weights $\{B_i\}_{i=0}^{n-1} \in \mathbb{Z}^{d_1 \times d_2}$, we aim to compute ciphertext results $\{C_i\}_{i=0}^{n-1}$ where $C_i = A_i \times B_i$. Each ciphertext can pack $N$ elements.

In the original BOLT [24] packing method, each matrix $A_i$ is packed column-wise, with a single ciphertext holding $\frac{N}{t}$ columns. To accumulate results across columns, $\frac{N}{t} - 1 = O\left(\frac{N}{t}\right)$ rotations are required per ciphertext. For all $n$ matrix multiplications, the total number of ciphertexts is $O(\frac{ntd_1}{N})$, leading to an overall rotation cost of $O\left(\frac{N}{t} \cdot \frac{ntd_1}{N}\right) = O(nd_1)$.

In contrast, our batch MatMul packing method packs columns from all $A_i$ matrices, allowing each ciphertext to store $\frac{N}{nt}$ columns (with each column containing $nt$ entries from $A_i{}_{i=0}^{n-1}$). This reduces the required rotations per ciphertext to $O\left(\frac{N}{nt}\right)$. While the total number of ciphertexts remains $O\left(\frac{ntd_1}{N}\right)$, the total rotation complexity drops to $O\left(\frac{ntd_1}{N} \cdot \frac{N}{nt}\right) = O(d_1)$.

**Integrate Baby-step Giant-step (BSGS) strategy into Batch MatMul packing**. The BSGS algorithm is commonly used for ciphertext-plaintext MatMul to reduce HE rotations [24], decomposing rotations into local (baby-step) and global (giant-step) phases. This reduces the number of rotations per-MatMul from $O(d_1)$ to $O\left(\sqrt{\frac{td_1d_2}{N}}\right)$. The total number of rotations is $O\left(n\sqrt{\frac{td_1d_2}{N}}\right)$. By combining BSGS with our batch packing, the effective parallel token dimension increases from $t$ to $nt$, since columns from all $n$ matrices are processed in parallel. This reduces the overall rotation complexity to $O\left(\sqrt{\frac{ntd_1d_2}{N}}\right)$, achieving a $\sqrt{n}\times$ improvement compared to applying BSGS independently to each matrix multiplication.

# E  Ablation Study on $t$ and Sequence Length

Figure 13 presents the ablation study on the impact of $t$ and input sequence length for both OLMoE and QWenMoE models. Consistent with Section 5.4, increasing either $t$ or the sequence length improves accuracy. Additionally, QWenMoE exhibits better load balancing, resulting in smaller accuracy degradation even with shorter sequences.

