# OpenReview forum: "CryptoMoE: Privacy-Preserving and Scalable Mixture of Experts Inference via Balanced Expert Routing"
_NeurIPS.cc/2025/Conference — NeurIPS 2025 poster_

### Official Review · Reviewer_d2x4 · 2025-06-04

**Clarity:** 2
**Significance:** 2
**Originality:** 2
**Rating:** 3
**Confidence:** 5

**Summary:**

This paper introduces CryptoMoE, the first cryptographic framework that enables privacy-preserving inference for Mixture-of-Experts (MoE) large language models. While prior private inference systems focus on dense architectures, CryptoMoE innovatively addresses the privacy risk posed by dynamic expert routing in MoE layers, which are highly input-dependent and thus leak-prone. The authors propose a novel balanced expert routing protocol and corresponding secure dispatch and combine operations under hybrid HE/MPC.

**Questions:**

* Have you benchmarked your Batch MatMul protocol against Nexus or CipherGPT under the same threading and HE parameter settings?

* Can you provide or commit to releasing code to allow independent verification?

**Ethical Concerns:**

["NO or VERY MINOR ethics concerns only"]

**Final Justification:**

After reading the author's rebuttal, I gave this rating.

**Limitations:**

* Given the significant performance improvements claimed over prior secure inference work, the lack of code release or detailed hardware configurations could hinder validation and adoption.

* While the paper acknowledges WAN latency and short-sequence inefficiencies, it omits important concerns around reproducibility, assumptions on the adversarial model, and multi-party scalability.

**Paper Formatting Concerns:**

* Inconsistent notation: For example, Figure 4 consistently uses “Top-t”, while the main text refers to the same concept as “top-t” in line 55 and “top-k” in line 312. It would improve clarity and coherence to adopt a consistent terminology throughout the paper.

* The reported speedup factors (e.g., 3×, 2×) in Figure 8 are unclear with respect to their reference baseline. Additionally, since the figure distinguishes between LAN and WAN, the caption should use the plural form settings to reflect the presence of two distinct evaluation environments.

**Quality:**

2

**Strengths And Weaknesses:**

**Strengths**
* First to tackle private MoE inference under secure computation. The routing problem is well-motivated and practically significant.
* Demonstrates strong empirical results, retaining accuracy while substantially improving efficiency compared to both insecure and dense baselines.

**Weaknesses**
* The matrix multiplication only compares the BOLT w/o BSGS version. Compared with the SOTA bumblebee[1] and the more suitable nexus [2] and ciphergpt [3] for batch matrix multiplications, there is no comparison, lacking novelty and effectiveness.
* Under 48 threads, the inference performance shown in the paper far exceeds that of sota's work [1,2,3], and no open source link is provided. Perhaps I need to check the code to eliminate my concerns.

**Reference**

[1] BumbleBee: Secure Two-party Inference Framework for Large Transformers

https://www.ndss-symposium.org/wp-content/uploads/2025-57-paper.pdf

[2] Secure Transformer Inference Made Non-interactive

https://www.ndss-symposium.org/wp-content/uploads/2025-868-paper.pdf

[3] CipherGPT: Secure Two-Party GPT Inference

https://eprint.iacr.org/2023/1147.pdf

---

> ### Author Rebuttal · Authors · 2025-07-31
>
> We thank Reviewer d2x4 for your valuable and detailed feedback!  Below we list our responses to each of the comments:
>
> -----
>
> **[To Q1 & W1: Comparison with BOLT (BSGS), Bumblebee, Nexus and CipherGPT on batch MatMul]**
>
> Our batch MatMul packing strategy is **fully compatible with the Baby-Step Giant-Step (BSGS)** algorithm. As analyzed in **Appendix E**, we provide a detailed complexity analysis for integrating BSGS with our batch packing method. Specifically, BSGS combined with batch packing reduces the number of required rotations from $O(n\sqrt{td_1 d_2/N})$ to $O(\sqrt{ntd_1 d_2/N})$, yielding a **$\sqrt{n} \times$ reduction** in rotation complexity. In our implementation, we **consistently apply the BSGS optimization**. For clarity, Figure 6 in our paper reports results **without** BSGS, but Appendix E covers both settings.
>
> In the table below, we compare our batch MatMul protocol against existing approaches, including BOLT [r24], NEXUS [r37], and CipherGPT (**VOLE-based**) [r44]. All evaluations are conducted on a machine equipped with an **Intel Xeon Platinum 8468 CPU (48 cores @ 2.1GHz, 64 threads)** under a LAN setting. Given ciphertext matrices $\\{A_i\\}\_{i=0}^{n-1} \in \mathbb{Z}^{t \times d_1}$ and plaintext weights $\\{B_i\\}_{i=0}^{n-1} \in \mathbb{Z}^{d_1 \times d_2}$, we compute $C_i = A_i \times B_i$ for $n = 64$, $t = 24$, $d_1 = d_2 = 2048$. These dimensions are derived from the DeepSeekMoE-2.8B/16.4B models.
>
> We follow the HE parameter settings used in Bumblebee, namely: $N = 8192$, $\log q = 114$, and $\log p = 64$, where $N$ is the polynomial degree, and $q$, $p$ are the ciphertext and plaintext moduli, respectively. For secret sharing, we adopt a 64-bit ring.
>
> | Method    | Comm. (GB) | \#Rotation (Key-switch) | Latency(s) |
> | --------- | ---------- | ----------------------- | ---------- |
> | Bolt      | 0.482      | 15360                   | 62         |
> | Nexus     | 0.482      | 8388480                 | 3216       |
> | CipherGPT | 106.5      | /                       | 2421       |
> | Ours      | 0.482      | 1024                    | 21         |
>
> The communication cost and computational latency for CipherGPT are estimated following their paper, as their code is not open-source. It is important to note that the MatMul protocols proposed by CipherGPT and Nexus are optimized for scenarios involving **different ciphertext input matrices $A_i$ multiplied by the same plaintext weight matrix $B$**. However, in the batch MatMul of the MoE block, **different experts process distinct ciphertext tokens with distinct weight matrices**. This means we cannot leverage the batch amortizable properties that CipherGPT and Nexus offer, resulting in their method incurring higher latency.
>
> The results show that our batch matrix multiplication protocol, **specifically optimized for MoE layers**, performs significantly better than the methods discussed above.
>
> ----
>
> **[To Q2 & W2 & Limitation 1: Concerns about performance and reproducibility]**
> We clarify that the reported inference time and communication cost in **Table 3** are **amortized per-token costs**, i.e., the total cost divided by the number of input tokens. **When considering this**, our results are consistent with prior works such as BOLT [r24] and Bumblebee [r31]. We include a table below summarizing the per-token cost across BOLT, Bumblebee and NEXUS for clarity.
>
> | Work        | Model                  | Bandwidth (Gbps) | Ping latency (ms) | Latency (min/token) | Comm. (GB/token) |
> | ----------- | ---------------------- | ---------------- | ----------------- | ------------------- | ---------------- |
> | Ours        | DeepSeekMoE 2.8B/16.4B | 3                | 0.2               | 0.76                | 2.46             |
> | Ours        | OLMoE 1.3B/6.9B        | 3                | 0.2               | 0.36                | 1.31             |
> | Bolt        | BERT-base              | 3                | 0.8               | 0.024               | 0.47             |
> | BumbleBee   | LLaMA-7B               | 1                | 0.5               | 1.73                | 0.85             |
> | BumbleBee   | BERT-large             | 1                | 0.5               | 0.052               | 0.16             |
> | Nexus (CPU) | Llama-3-8B (batch32)   | 3                | 0.8               | 2.27                | /                |
> | Nexus (CPU) | BERT-base (batch32)    | 3                | 0.8               | 0.11                | 0.0013           |
>
> We would like to emphasize that prior works such as BOLT and Bumblebee focus on supporting dense models (e.g., BERT, LLaMA-1). In contrast, our core contribution lies in **proposing the first framework that supports privacy-preserving inference for MoE models**. For other components, such as RMSNorm and attention, we **directly adopt the protocols from prior works BOLT and Bumblebee**. Specifically, for matrix multiplications in linear layers, we leverage BOLT’s protocol; for nonlinear operations, including RMSNorm, Softmax, and SiLU, we employ the protocols proposed in Bumblebee. We acknowledge that these aspects are not the primary focus of our work. In addition, the communication overhead of MoE models is relatively higher than that of dense models, primarily due to the additional communication introduced by the dispatch and combine protocols. However, MoE models reduce the overall computation per expert, leading to lower latency and thus offering better efficiency in terms of inference time.
>
> **As for our HE/MPC parameter configuration**, we follow the settings in Bumblebee: we use $N = 8192$, $\log q = 114$, and $\log p = 64$, where $N$ is the polynomial degree, and $q$, $p$ are the ciphertext and plaintext moduli, respectively. For secret sharing, we adopt a 64-bit ring. We will include these details in the revised version to improve clarity.
>
> **Regarding code availability**, we understand that an open-source release is vital for independent verification and fostering adoption. Our current code framework is built upon **Secretflow SPU**, and we are actively in the process of organizing and refining it for public release. We are fully committed to releasing a reference implementation of our code once our paper is accepted.
>
> -----
>
> **[To Limitation 2: Assumptions on the Adversarial Model and Multi-Party Scalability]**
>
> Our work focuses on the **two-party setting under the semi-honest threat model**, which is a common and practical assumption in privacy-preserving machine learning. Extending the framework to a malicious adversarial model or to a multi-party setting falls outside the current scope of our study. Nevertheless, both directions are important and promising. As noted in BOLT, **designing protocols that are secure against malicious adversaries is far from trivial**. Similarly, supporting multi-party computation would require **replacing the underlying MPC primitives we currently employ**, which we also consider a valuable avenue for future work. We will add a discussion of these points in the final version to enhance completeness.
>
> ------
>
> **[To Paper Formatting Concerns]**
>
> Thank you for your careful review and helpful suggestions.
>
> Regarding the notation inconsistency between "top-t" and "top-k", we clarify that our intention was to use **"top-k"** consistently to refer to the general protocol for selecting the top-$k$ elements (e.g., $\Pi_{\text{top}k}$). The occurrence of "top-t" in line 55 was meant to emphasize that **each expert receives $t$ tokens** during dispatch. We will standardize the notation to **"top-k"** throughout the paper and explicitly state **$k = t$** in relevant contexts to eliminate ambiguity.
>
> For Figure 8, the **reported speedup factors (e.g., 3×, 2×)** refer to the performance gain of our proposed method compared to the **dense baseline**. The figure caption will be revised to use the plural form "settings" in the final version.
>
> We will revise the final version to improve the consistency and clarity of notation and figures to avoid misunderstandings.
>
> -----
>
> **References**
>
> - \[r24, r31, r37, r44] represent the 24th, 31th, 37th, 44th references cited in our original submission.
>
> ----
>
> **We sincerely appreciate the detailed advice from reviewer d2x4! We take all your suggestions into account in our final version. Please let us know if you have any further questions. If you feel that the above improvements help clear up your doubts and further improve our work, you can kindly consider a re-evaluation, thank you!**

---

> > ### Comment · Reviewer_d2x4 · 2025-08-02
> >
> > Thank you for the detailed responses. While I understand the policy constraints, I still believe that code availability is important for reproducibility. However, I acknowledge the authors’ clarifications and have no further questions.

---

> > > ### Author Response · Authors · 2025-08-03
> > >
> > > Thank you for your understanding and for highlighting the importance of reproducibility. We completely agree with you.
> > >
> > > As clarified earlier, due to policy constraints (pending institutional approval and patent filing), we are temporarily unable to release the code during the review phase. However, we are firmly committed to releasing the codebase upon paper acceptance and release, in line with community best practices. This includes both private inference protocols and evaluation scripts to reproduce both accuracy and inference cost results.
> > >
> > > We appreciate your acknowledgment and thoughtful comments, and we hope our response reassures you and the Aera Chairs that we intend to support reproducibility to the fullest extent once external constraints are lifted.

---

### Official Review · Reviewer_8zVq · 2025-06-23

**Clarity:** 2
**Significance:** 3
**Originality:** 3
**Rating:** 4
**Confidence:** 3

**Summary:**

The authors mainly address the inefficiency of performing MoE securely. To reduce the number of tokens dispatched to experts, they select the top-k tokens for each expert based on confidence scores. For efficient execution, they propose new protocols for dispatching, combining, and batch matrix multiplication. In the top-k selection, they reduce complexity by avoiding secure swaps compared to the protocol used in CipherPrune. For the combine protocol, they efficiently aggregate the results by reusing the one-hot matrix. Finally, by applying column-packing to the combined input matrix, they reduce the number of required rotations in batch matrix multiplication. Experimental results demonstrate that the proposed method achieves a 2×–4× improvement in efficiency.

**Questions:**

Please refer to the Weaknesses section.

**Ethical Concerns:**

["NO or VERY MINOR ethics concerns only"]

**Final Justification:**

The problem is well-specified, and the proposed method is novel. Therefore, I keep my score as 4: borderline accept.

**Limitations:**

The authors adequately discuss the limitations in Section 6.

**Paper Formatting Concerns:**

No formatting issues were found.

**Quality:**

3

**Strengths And Weaknesses:**

### Strengths
* This work is the first to identify the inefficiencies that can arise when performing MoE securely.
* The authors achieved a 2×–4× speedup by using confidence-based selection to mitigate the inefficiencies in naively performing MoE securely.
* The proposed protocol is well-designed for the given problem.
* The effectiveness of the proposed method was validated on large language models.

---

### Weaknesses
* In Section 4.4, Bolt utilizes the Baby-Step Giant-Step (BSGS) algorithm to reduce computational complexity when the polynomial dimension is large, but this aspect has not been considered in the paper. It is necessary to analyze whether the proposed method is also compatible with BSGS. If not, a comparison with Bolt incorporating BSGS should be provided.
* Additionally, prior to the MoE operation, the ciphertexts are likely structured in a column-packed format without sample mixing. With the proposed method, repacking of column-packed ciphertexts would be necessary, but the analysis of this additional cost appears to be missing.
* The description of the experimental setup for the end-to-end comparison is insufficient. The experiments were conducted on a very large model, yet the reported communication cost and speed seem relatively low. It is unclear whether only the MoE component was computed securely while the remaining parts were processed in plaintext. If the entire model was executed securely, it would be helpful to provide detailed explanations on how non-linear operations and matrix multiplications in the attention mechanism were handled, including which methods or protocols were employed.

Typos
* "Threat Model" is miswritten as "Thread Model" in line 99.

---

> ### Author Rebuttal · Authors · 2025-07-31
>
> We thank Reviewer 8zVq for your support and for the professional and detailed feedback! Below is our response.
>
> -----
>
> **[To W1: Compatibility with BOLT BSGS]**
>
> Our batch MatMul packing strategy is **compatible with the Baby-Step Giant-Step (BSGS)** algorithm. As detailed in **Appendix E** of our paper, we provide a complexity analysis of integrating BSGS with our packing. Specifically, BSGS combined with batch packing reduces the number of required rotations from $O(n\sqrt{td_1 d_2/N})$ to $O(\sqrt{ntd_1 d_2/N})$, resulting in a $\sqrt{n} \times$ improvement in rotation complexity.
>
> -----
>
> **[To W2: Cost of repacking for batch MatMul packing]**
>
> We would like to clarify that both the input and output of our dispatch and combine protocols are in **secret share (SS) form**, not HE ciphertext. As shown in Algorithms 1 and 2 in Appendix C, under the SS form, the repacking operations can be performed **locally and in plaintext**, incurring **negligible cost**. Thus, our batch MatMul format does **not** introduce extra packing cost in the HE computation.
>
> -----
>
> **[To W3: Setup details for end-to-end evaluation]**
>
> All components of our Transformer model—including RMSNorm, attention, and MoE blocks—are evaluated **securely under our HE/MPC hybrid framework**, ensuring **end-to-end privacy**. The reported inference time and communication cost in **Table 3** are **amortized per-token costs**, i.e., total cost divided by the number of input tokens. When considering this, our results are consistent with prior works such as BOLT [r24] and Bumblebee [r31]. We include a table below summarizing the per-token cost across BOLT and Bumblebee for clarity.
>
> | Work            | Model                  | Bandwidth (Gbps) | Ping latency (ms) | Latency (min/token) | Comm. (GB/token) |
> | --------------- | ---------------------- | ---------------- | ----------------- | ------------------- | ---------------- |
> | Ours            | DeepSeekMoE 2.8B/16.4B | 3                | 0.2               | 0.76                | 2.46             |
> | Ours            | OLMoE 1.3B/6.9B        | 3                | 0.2               | 0.36                | 1.31             |
> | Bolt [r24]      | BERT-base (0.1B)       | 3                | 0.8               | 0.024               | 0.47             |
> | BumbleBee [r31] | LLaMA-7B               | 1                | 0.5               | 1.73                | 0.85             |
> | BumbleBee [r31] | BERT-large (0.3B)      | 1                | 0.5               | 0.052               | 0.16             |
>
> We would like to emphasize that prior works such as BOLT and Bumblebee focus on supporting dense models (e.g., BERT, LLaMA-1). In contrast, our core contribution lies in **proposing the first framework that supports privacy-preserving inference for MoE models**. For other components, such as RMSNorm and attention, we **directly adopt the protocols from prior works BOLT and Bumblebee**. Specifically, for matrix multiplications in linear layers, we leverage BOLT’s protocol, and for nonlinear operations, including RMSNorm, Softmax, and SiLU, we employ the protocols proposed in Bumblebee. We acknowledge that these aspects are not the primary focus of our work.
>
> Furthermore, as shown in **Figure 8**, when the total number of input tokens is relatively small (e.g., typically under 256 tokens), **the MoE layer becomes the dominant bottleneck** in both LAN and WAN settings. This observation further highlights our key contribution: **enabling efficient private inference for the MoE module**.
>
> We will incorporate this clarification in the revised version to avoid potential confusion. We hope this addresses your concerns and provides a clearer understanding of the scope and focus of our design and performance contributions.
>
> **[To Typos]**
>
> Thank you for pointing out that "Threat Model" was misspelled as "Thread Model".  We will correct this in our final version.
>
> **References**
>
> - \[r24, r31] represent the 24th, 31th references cited in our original submission.
>
> ----
>
> **Last, thanks so much for helping us improve this work through your professional perspective!**

---

### Official Review · Reviewer_SQLk · 2025-07-02

**Clarity:** 3
**Significance:** 3
**Originality:** 3
**Rating:** 5
**Confidence:** 5

**Summary:**

This work propose an framework to securely and efficiently process the MoE layer in modern LLM models via HE+2PC. With proposed confidence-aware dispatching, it reduced the computation and communicated overhead compared to fully-protected baseline-dense. With multiple-token batched into single ciphertext, it reduces the rotation complexity.

**Questions:**

1.When performing the evaluation, at which point the data communicate back to client? I think this part should be added with some figures or algorithms to clearly show the overall flow. For example, I guess the highest communication overhead of baseline-dense comes from the   all-tokens sent to each expert, which led to tremendous intermediate results. It would be better to illustrate this in details.

2.I think a detailed configurations for HE should be included in setup part.

3.To better understanding the dispatch protocols, I think I need authors to help me understand the functions of m secret-shared tokens. By splitting the tokens into m shares, it seems that these share could compute with each expert in weighted-sum manner. Does this for accuracy improve? I am not sure why the tokens are secret-shared?

I hope authors could address my concerns, thanks!

**Ethical Concerns:**

["NO or VERY MINOR ethics concerns only"]

**Final Justification:**

Considering author's detailed rebuttal, I will keep my score for recommendation.

**Limitations:**

yes

**Paper Formatting Concerns:**

I see one issue in 1st para of section 2: "1 {P}...", and guess this is a typo.

The overall writing and formatting is very good and well-formatted.

**Quality:**

3

**Strengths And Weaknesses:**

Strengths:
1. This is an very interesting research problem and still remains un-explored, in which sensitive info requiring protection.
2. With a index-obfuscating style method, it balance the privacy leakage and computation efficiency.

Weakness:
1. Missing some HE parameter setting, I think it is important for readers to know due to it is highly related with HE computation latency.
2. Missing a detailed general HE computation protocol used in evaluation, including the computation and communication illustration together.

---

> ### Author Rebuttal · Authors · 2025-07-31
>
> Thank you very much for your strong support and the thorough and creative comments! We also appreciate your evaluation of our work as “a very interesting research problem.” Please find our responses to your questions below.
>
> ----
>
> **[To Q1: Details on the framework’s communication flow]**
>
> Our framework adopts the **hybrid HE/MPC scheme**, a widely used approach in recent secure ML research [r24, r25, r26, r27, r28]. In this scheme, Homomorphic Encryption (HE) is used to efficiently handle linear layers, while Multi-Party Computation (MPC)—including secret sharing (SS) and Oblivious Transfer (OT)—is employed for non-linear operations. This hybrid design involves communication in the following two main cases:
>
> - **HE–SS Conversion**: Communication arises at the interface between linear and non-linear layers, where HE ciphertexts must be converted to/from SS form.
> - **MPC Protocol Execution**: Communication is inherent to executing MPC protocols for non-linear functions. In particular, Oblivious Transfer (OT), a core building block for non-linear computation, requires interactive communication between the client and the server.
>
> Figures 4 and 5 in our manuscript illustrate how HE and MPC protocols are integrated into our dispatch and combine steps. During expert computation, the SiLU activation is performed via MPC, while the surrounding linear matrix multiplications are handled by HE.
>
> Your analysis of the dense baseline is correct. The dense baseline involves substantial communication cost because **each expert processes all $m$ tokens**, producing a large volume of intermediate results. These results then require **conversion between HE and SS representations**, causing significant HE ciphertext transmission. Additionally, the dense computation increases both linear and non-linear processing and communication costs.
>
> ------
>
> **[To Q2 & W1: HE parameter settings]**
>
> We follow the configuration of Bumblebee for HE parameters and set $N=8192$, $\log q=114$, and $\log p=64$, where $N$ is the polynomial degree, and $q$, $p$ are the ciphertext and plaintext moduli, respectively. For secret sharing, we adopt a 64-bit ring.
>
> ------
>
> **[To Q3: Explanation on the dispatch protocol]**
>
> The primary function of secret sharing in our protocol is to ensure data privacy: neither the client nor the server can access the full data independently. As shown in Figure 4, we process $m$ tokens of hidden dimension $d$, resulting in data of shape $m \times d$. This data is converted into secret shares to maintain privacy. Throughout our protocol, all intermediate results are represented in secret-shared form.
>
> Regarding your suggestion to pre-split tokens into $m$ shares for weighted-sum computation at each expert—this is a very insightful idea. However, due to two critical constraints:
>
> - Each expert applies a SiLU non-linear activation, which makes pre-aggregation of experts' weights infeasible.
> - The expert weights must also remain private, and thus cannot be revealed or used to compute a public weighted sum.
>
> As a result, we must design a secure $\Pi_{\text{dispatch}}$ and $\Pi_{\text{combine}}$ protocol based on secret sharing. Your idea would be well-suited to **purely linear pipelines**, where pre-aggregation of weights can be applied in a reparameterization-style approach as in [a1].
>
> We hope this clarifies the necessity and role of secret-shared tokens in our protocol.
>
> ------
>
> **[To W2: General HE computation protocol]**
>
> For linear layer computations, we adopt the MatMul protocol described in Section 4.1.1 of BOLT [r24]. Furthermore, for matrix multiplication within the MoE block, we apply an additional **batch optimization** on top of the BOLT protocol. We will incorporate a more detailed explanation of the BOLT-based HE protocol in the appendix—thank you for the suggestion!
>
> ------
>
> **[To formatting concerns]**
>
> Thank you for pointing out the formatting of $\bf{1}\\{\mathcal{P}\\}$ in the first paragraph of Section 2. We use $\bf{1}\\{\mathcal{P}\\}$ to denote an indicator function, which returns 1 when condition $\mathcal{P}$ is true and 0 otherwise. For instance, ${\bf{1}}\\{x == 0\\}$ evaluates to 1 if $x = 0$, and 0 otherwise. We will revise the notation for better clarity and avoid confusion in the final version.
>
> ------
>
> **References**
>
> - [r24, r25, r26, r27, r28] correspond to references 24 through 28 in our original submission.
> - [a1] Ding, Xiaohan, et al. *RepVGG: Making VGG-style ConvNets Great Again*. In: Proceedings of the IEEE/CVF Conference on Computer Vision and Pattern Recognition (CVPR), 2021.
>
> ----
>
> **Thank you again for your strong support and the valuable comments, which have strengthened and made our paper more robust! We will add all of these discussions to our final version.**

---

> > ### Comment · Reviewer_SQLk · 2025-08-06
> > **Response**
> >
> > Thanks for rebuttal. I have no more concerns.

---

### Official Review · Reviewer_NVGY · 2025-07-03

**Clarity:** 3
**Significance:** 3
**Originality:** 3
**Rating:** 4
**Confidence:** 4

**Summary:**

This paper introduces CryptoMoE, the first cryptographic framework for private and efficient inference on Mixture-of-Experts (MoE) LLMs. MoE architectures dynamically route input tokens to different experts based on content, which can unintentionally leak sensitive information. CryptoMoE addresses this challenge by introducing Inference-Time Balanced Expert Routing—a privacy-preserving mechanism that ensures expert activation patterns are input-independent. To maintain accuracy, it uses a confidence-aware token selection and develops secure protocols for expert dispatch and combination. Additionally, it introduces a batch matrix multiplication protocol to improve the efficiency of homomorphic encryption operations. Evaluations on MoE models show 3× reduction in latency and communication cost, outperforming both dense and pruned baselines.

**Questions:**

1. Could you clarify how to implement the dense baseline and what protocols are used to achieve it?

2. Could you explain why CryptoMoE outperforms the dense baseline, considering that multiple expensive non-linear operations like onehot and topk are involved in CryptoMoE?

**Ethical Concerns:**

["NO or VERY MINOR ethics concerns only"]

**Quality:**

3

**Strengths And Weaknesses:**

Strengths

+ Support private inference for MoE-based LLMs
+ Protect expert routing information with private dispatch and combination
+ Efficient protocols for batched matrix multiplications
+ 2.8–3.5× latency reduction and 2.9–4.3× communication reduction

Weaknesses

- The protocols, including dispatch and combination, are designed using existing building blocks.
- It is unclear how to implement the dense baseline and what protocols are used to achieve it.

---

> ### Author Rebuttal · Authors · 2025-07-31
>
> Thank you very much for your support and constructive comments, which have greatly helped us improve our work! See below for the answers to your questions and comments.
>
> -----
>
> **[To Q1 & W2: Details on the dense baseline implementation]**
>
> We provide a detailed implementation of the **dense baseline** below, which complements the brief description in Section 3 of our paper. The dense baseline evaluates **all experts for each token**, thereby fully hiding the routing information. The core steps are as follows:
>
> - **Routing and Score Matrix Construction:** After the gate routing step, we obtain the expert indices and scores $K, W \in \mathbb{Z}^{m \times k}$, where $m$ is the number of tokens and $k$ is the number of selected experts per token.  We then convert $K$ into a one-hot representation as follows: $K_{\text{onehot}} = \text{onehot}(K) \in \mathbb{Z}^{m \times k \times n}$, where $n$ is the total number of experts. Within the $n$-dimension of $K_{\text{onehot}}$, only the entries corresponding to the selected expert indices are set to 1, and all others are 0. We then compute the expert score matrix: $S = W \times K_{\text{onehot}} \in \mathbb{Z}^{m \times n}$, where the multiplication reduces along the $k$-dimension, and $m$ remains the parallel axis. Each row $S_i$ reflects the expert scores for the $i$-th token: the positions corresponding to selected experts contain their routing scores, while the rest are 0.
> - **Expert Computation:**  Each expert $E_i$ computes on **all** tokens: $y_{E_i} = E_i(x) \in \mathbb{Z}^{m \times d}$, aggregating all expert outputs gives: $y_E \in \mathbb{Z}^{n \times m \times d}$.
> - **Combination:**  Finally, we compute the output of the MoE layer as: $y = S \times y_E \in \mathbb{Z}^{m \times d}$, where the multiplication reduces over the expert dimension $n$, and $m$ remains the parallel dimension. This operation essentially computes a weighted sum over expert outputs for each token.
>
> The corresponding privacy-preserving inference algorithm is described below:
>
> **Algorithm: Dense Baseline**
>  **Input:** Secret-shared token embeddings $[\\![x]\\!] \in \mathbb{Z}^{m \times d}$, with $m$ tokens and $d$-dimensional embeddings; $k$ is the number of selected experts.
>  **Output:** Secret shares of MoE output $[\\![y]\\!] \in \mathbb{Z}^{m \times d}$.
>
> 1. Use $\Pi_{\text{matmul}}$, $\Pi_{\text{softmax}}$, and $\Pi_{\text{topk}}$ to perform secure routing and obtain $[\\![K]\\!], [\\![W]\\!] \in \mathbb{Z}^{m \times k}$.
> 2. Compute outputs for all experts: $[\\![y_{E_i}]\\!] = E_i([\\![x]\\!]) \in \mathbb{Z}^{m \times d}$ for $i \in [0, n-1]$; reshape to $[\\![y_E]\\!] \in \mathbb{Z}^{n \times m \times d}$.
> 3. Apply $\Pi_{\text{onehot}}([\\![K]\\!], c=n)$ to obtain one-hot routing matrix $[\\![\text{onehot}(K)]\\!]^B \in \mathbb{Z}^{m \times k \times n}$.
> 4. Compute expert score matrix:
>    $[\\![S]\\!] \leftarrow \Pi_{\text{matmul}}([\\![W]\\!], [\\![\text{onehot}(K)]\\!]^B) \in \mathbb{Z}^{m \times n}$.
> 5. Compute MoE output:
>    $[\\![y]\\!] \leftarrow \Pi_{\text{matmul}}([\\![S]\\!], [\\![y_E]\\!]) \in \mathbb{Z}^{m \times d}$.
>
> ------
>
> **[To Q2: Why CryptoMoE outperforms the dense baseline]**
>
> As shown in the ablation study in **Figure 8**, CryptoMoE significantly reduces the cost of expert computation compared to the dense baseline. In the dense baseline, each expert processes all $m$ tokens, leading to a high computational cost. In contrast, $\text{CryptoMoE}\_{t=2.0}$ reduces this to $2mk/n$ tokens per expert. Taking DeepSeekMoE-2.8B/16.4B as an example, with $m=128$, $k=6$, and $n=64$, each expert processes only $2 \times 128 \times 6 / 64 = 24$ tokens, whereas the dense baseline requires computing all $128$ tokens. CryptoMoE saves **over $500\\%$** of the expert computation cost, which is the main bottleneck in the MoE block. Although CryptoMoE introduces additional operations such as Top-K and one-hot encoding, these account for **only 18%** of the total LAN latency, as shown in Figure 8. Therefore, the savings in the expert layers far outweigh the cost in $\Pi_{\text{dispatch}}$ and $\Pi_{\text{combine}}$.
>
> ------
>
> **[To W1: Dispatch and combination protocols are designed using existing building blocks.]**
>
> Our $\Pi_{\text{dispatch}}$ and $\Pi_{\text{combine}}$ protocols are built upon well-established cryptographic primitives [r46]. **Our key contributions lie in the token manipulation strategy designed specifically for secure MoE inference**, which enables secure token-to-expert assignment and result aggregation **without leaking any sensitive information**. The underlying cryptographic primitives, such as comparison protocols, are not our focus.
>
> To highlight the novelty and effectiveness of our $\Pi_{\text{dispatch}}$ and $\Pi_{\text{combine}}$ protocols, we compare them with the protocol used in CipherPrune [r22]. Prior works [r22, r24] rely on MPC-based sorting that directly swaps **high-dimensional token embeddings**, resulting in significant communication and latency cost. This challenge is especially pronounced in MoE blocks, where sorting is required in both the dispatch and combine phases for each expert.
>
> In contrast, **CryptoMoE eliminates the need for direct manipulation of embeddings** by decoupling token indices from embeddings. This design significantly reduces the communication cost. The table below compares CryptoMoE and CipherPrune on a single MoE block of DeepSeekMoE under a LAN setting, where $m$, $n$, $k$, $t$, and $d$ represent the number of input tokens, number of experts, number of selected experts, dispatched tokens per expert, and embedding dimension, respectively. The evaluation is conducted using the following parameters: $m=128$, $n=64$, $k=6$, $t=12$, $d=2048$.
>
> | Method      | Communication complexity | Dispatch Communication (GB) | Dispatch latency (s) | Combine Communication (GB) | Combine latency (s) |
> | ----------- | ------------------------ | --------------------------- | -------------------- | -------------------------- | ------------------- |
> | CipherPrune | $nkmtd$                  | 9.67                        | 90.8                 | 6.44                       | 52.6                |
> | Ours        | $nkm\log(km)+ntm$        | 0.89                        | 34.4                 | 0.72                       | 3.0                 |
>
> More detailed analysis is provided in **Appendix D** in our paper.
>
> ------
>
> **References**
>
> - \[r22, r24, r46] represent the 22th, 24th, 46th references cited in our original submission.
>
> ----
>
> **We sincerely appreciate the professional, detailed advice from reviewer NVGY! We hope this response fully addresses your concerns.**

---

### Note · Authors · 2025-08-15

Dear Reviewers (NVGY, SQLk, 8zVq, d2x4) and Area Chairs,

We sincerely thank you for your time, thoughtful feedback, and constructive discussions throughout the review process. We are glad that our clarifications have addressed the raised concerns, and deeply appreciate that **all reviewers have kindly indicated they have no further questions**.

In particular, we are thankful for the discussions on our experimental comparisons and reproducibility. We would like to formally restate our **firm commitment to releasing the full implementation of CryptoMoE** upon acceptance and public release of the paper, once institutional approvals are cleared.

We will ensure that all promised clarifications, additional details (e.g., the complete HE MatMul protocol), and corrections are fully incorporated into the final camera-ready version.

**Once again, we extend our heartfelt thanks to all reviewers and the Area Chairs for your professional, constructive feedback, and the considerable time you have devoted to our work.**

---

### Decision · Program_Chairs · 2025-09-17

**Decision:**

Accept (poster)

**Comment:**

This paper introduces CryptoMoE, a novel framework for performing private and efficient inference on Mixture-of-Experts (MoE) Large Language Models. The core challenge addressed is the potential for information leakage from the dynamic routing mechanism in MoE models, where input tokens are sent to different "expert" sub-networks depending on their information. The authors propose a cryptographic system to hide which experts are being used for a given input. Their method includes a balanced expert routing strategy and new protocols for secure data handling. According to the experimental results, CryptoMoE achieves a multiple factor reduction in both latency and communication overhead compared to a standard dense model (without losing accuracy).

The reviews for this paper are borderline, with final ratings ranging from "Accept" to "Borderline reject". On the positive side, all reviewers agree that the paper tackles a novel and important problem, being the first to address private inference for MoE models. They also acknowledge the strong empirical results and the well-designed approach. However, significant concerns make the paper's acceptance questionable. Multiple reviewers pointed to a lack of clarity in the experimental setup, requesting more details on baselines and cryptographic parameters. There were also concerns about insufficient comparison to other state-of-the-art methods, which was partly resolved during the rebuttal period. There is one major remaining concern about the lack of publicly available code which hinders the community from reproducing the result (especially given the impressive performance reported).